# Adversarial Risk and Robustness: General Definitions and Implications for the Uniform Distribution

**Dimitrios I. Diochnos***
University of Virginia
diochnos@virginia.edu

**Saeed Mahloujifar***
University of Virginia
saeed@virginia.edu

**Mohammad Mahmoody**†
University of Virginia
mohammad@virginia.edu

## Abstract

We study adversarial perturbations when the instances are uniformly distributed over $\{0,1\}^n$. We study both "inherent" bounds that apply to any problem and any classifier for such a problem as well as bounds that apply to specific problems and specific hypothesis classes.

As the current literature contains multiple definitions of adversarial risk and robustness, we start by giving a taxonomy for these definitions based on their direct goals; we identify one of them as the one guaranteeing misclassification by pushing the instances to the *error region*. We then study some classic algorithms for learning monotone conjunctions and compare their adversarial robustness under different definitions by attacking the hypotheses using instances drawn from the uniform distribution. We observe that sometimes these definitions lead to *significantly different* bounds. Thus, this study advocates for the use of the error-region definition, even though other definitions, in other contexts with context-dependent assumptions, may coincide with the error-region definition.

Using the error-region definition of adversarial perturbations, we then study *inherent* bounds on risk and robustness of *any* classifier for *any* classification problem whose instances are uniformly distributed over $\{0,1\}^n$. Using the isoperimetric inequality for the Boolean hypercube, we show that for initial error $0.01$, there always exists an adversarial perturbation that changes $O(\sqrt{n})$ bits of the instances to increase the risk to $0.5$, making classifier's decisions meaningless. Furthermore, by also using the central limit theorem we show that when $n \to \infty$, at most $c \cdot \sqrt{n}$ bits of perturbations, for a universal constant $c < 1.17$, suffice for increasing the risk to $0.5$, and the same $c\sqrt{n}$ bits of perturbations *on average* suffice to increase the risk to $1$, hence bounding the robustness by $c \cdot \sqrt{n}$.

## 1 Introduction

In recent years, modern machine learning tools (e.g., neural networks) have pushed to new heights the classification results on traditional datasets that are used as testbeds for various machine learning methods.[1] As a result, the properties of these methods have been put into further scrutiny. In particular, studying the *robustness* of the trained models in various adversarial contexts has gained special attention, leading to the active area of *adversarial* machine learning.

Within adversarial machine learning, one particular direction of research that has gained attention in recent years deals with the study of the so-called *adversarial perturbations* of the test instances. This line of work was particularly popularized, in part, by the work of Szegedy et al. [32] within

the context of deep learning classifiers, but the same problem can be asked for general classifiers as well. Briefly, when one is given a particular instance $x$ for classification, an adversarial perturbation $x'$ for that instance is a new instance with minimal changes in the features of $x$ so that the resulting perturbed instance $x'$ is misclassified by the classifier $h$. The perturbed instance $x'$ is commonly referred to as an *adversarial example* (for the classifier $h$). Adversarial machine learning has its roots at least as back as in [19, 24, 17]. However, the work of [32] revealed pairs of images that differed slightly so that a human eye could not identify any real differences between the two, and yet, contrary to what one would naturally expect, machine learning classifiers would predict different labels for the classifications of such pairs of instances. It is perhaps this striking resemblance to the human eye of the pairs of images that were provided in [32] that really gave this new push for intense investigations within the context of adversarial perturbations. Thus, a very intense line of work started, aiming to understand and explain the properties of machine learning classifiers on such adversarial perturbations; e.g., [15, 23, 2, 8, 20]. These attacks are also referred to as *evasion* attacks [25, 4, 15, 8, 36]. There is also work that aims at making the classifiers more robust under such attacks [27, 36], yet newer attacks of Carlini and Wagner [7] broke many proposed defenses.

**Our general goal.** In this work, we study barriers against robust classification of adversarial examples. We are particularly interested in foundational bounds that potentially apply to broad class of problems and distributions. One can study this question from the perspectives of both risk and robustness. In the case of risk, the adversary's goal is to increase the error probability of the classifier (e.g., to reach risk 0.5) by small perturbations of the instances, and in the case of robustness, we are interested in the *average* amount of perturbations needed for making the classifier always fail.

**Studying the uniform distribution.** We particularly study adversarial risk and robustness for learning problems where the input distribution is $U_n$ which is uniform over the hypercube $\{0,1\}^n$. We measure the cost of perturbations using the natural metric of Hamming distance. Namely, the distance between the original and perturbed instances $x, x' \in \{0,1\}^n$ is the number of locations that they are different. This class of distributions already include many learning problems of interest. So, by studying adversarial risk and robustness for such a natural distribution, we can immediately obtain results for a broad class of problems. We believe it is crucial to understand adversarial risk and robustness for natural distributions (e.g., $U_n$ uniform over the hypercube) and metrics (e.g., the Hamming distance) to develop a theory of adversarial risk and robustness that can ultimately shed light on the power and limitations of robust classification for practical data sets. Furthermore, natural distributions like $U_n$ model a broad class of learning problems directly; e.g., see [5, 28, 18, 30]. The hope is that understanding the limitations of robust learning for these basic natural distributions will ultimately shed light on challenges related to addressing broader problems of interest.

**Related work.** The work of Gilmer et al. [14] studied the above problem for the special case of input distributions that are uniform over unit spheres in dimension $n$. They showed that for any classification problem with such input distribution, so long as there is an initial constant error probability $\mu$, the robustness under the $\ell_2$ norm is at most $O(\sqrt{n})$. Fawzi et al. [11] studied the above question for Gaussian distributions in dimension $n$ and showed that when the input distribution has $\ell_2$ norm $\approx 1$, then by $\approx \sqrt{n}$ perturbations in $\ell_2$ norm, we can make the classifier *change its prediction* (but doing this does not guarantee that the perturbed instance $x'$ will be misclassified). Schmidt et al. [29] proved limits on robustness of classifying uniform instances by specific classifiers and using a definition based on "corrupted inputs" (see Section 2), while we are mainly interested in bounds that apply to any classifiers and guarantee misclassification of the adversarial inputs.

**Discussion.** Our results, like all other current provable bounds in the literature for adversarial risk and robustness only apply to specific distributions that do not cover the case of image distributions. These results, however, are first steps, and indicate similar phenomena (e.g., relation to isoperimetric inequalities). Thus, as pursued in [14], these works motivate a deeper study of such inequalities for real data sets. Finally, as discussed in [11], such theoretical attacks could *potentially* imply direct attacks on real data, *assuming* the existence of smooth generative models for latent vectors with theoretically nice distributions (such as Gaussian or uniform over the hypercube) into natural data.

## 1.1 Our Contribution and Results

As mentioned above, our main goal is to understand inherent barriers against robust classification of adversarial examples, and our focus is on the uniform distribution $U_n$ of instances. In order to achieve that goal, we both do a definitions study and prove technical limitation results.

**General definitions and a taxonomy.** As the current literature contains multiple definitions of adversarial risk and robustness, we start by giving a taxonomy for these definitions based on their direct goals. More specifically, suppose $x$ is an original instance that the adversary perturbs into a "close" instance $x'$. Suppose $h(x), h(x')$ are the predictions of the hypothesis $h(\cdot)$ and $c(x), c(x')$ are the true labels of $x, x'$ defined by the concept function $c(\cdot)$. To call $x'$ a successful "adversarial example", a natural definition would compare the predicted label $h(x')$ with some other "anticipated answer". However, what $h(x')$ is exactly compared to is where various definitions of adversarial examples diverge. We observe in Section 2 that the three possible definitions (based on comparing $h(x')$ with either of $h(x), c(x)$ or $c(x')$) lead to three different ways of defining adversarial risk and robustness. We then identify one of them (that compares $h(x)$ with $c(x')$) as the one guaranteeing misclassification by pushing the instances to the *error region*. We also discuss natural conditions under which these definitions coincide. However, these conditions do not hold *in general*.

**A comparative study through monotone conjunctions.** We next ask: how close/far are these definitions in settings where, e.g., the instances are drawn from the uniform distribution? To answer this question, we make a comparative study of adversarial risk and robustness for a particular case of learning monotone conjunctions under the uniform distribution $U_n$ (over $\{0,1\}^n$). A monotone conjunction $f$ is a function of the form $f = (x_{i_1} \wedge \cdots \wedge x_{i_k})$. This class of functions is perhaps one of the most natural and basic learning problems that are studied in computational learning theory as it encapsulates, in the most basic form, the class of functions that determine which features should be included as relevant for a prediction mechanism. For example, Valiant in [35] used this class of functions under $U_n$ to exemplify the framework of evolvability. We attack monotone conjunctions under $U_n$ in order to contrast different behavior of definitions of adversarial risk and robustness.

In Section 3, we show that previous definitions of robustness that are not based on the error region, lead to bounds that do *not* equate the bounds provided by the error-region approach. We do so by first deriving theorems that characterize the adversarial risk and robustness of a given hypothesis and a concept function under the uniform distribution. Subsequently, by performing experiments we show that, on average, hypotheses computed by two popular algorithms (FIND-S [22] and SWAP-PING ALGORITHM [35]) also exhibit the behavior that is predicted by the theorems. Estimating the (expected value of) the adversarial risk and robustness of hypotheses produced by *other* classic algorithms under specific distributions, or for other concept classes, is an interesting future work.

**Inherent bounds for any classification task under the uniform distribution.** Finally, after establishing further motivation to use the error-region definition as the default definition for studying adversarial examples in *general* settings, we turn into studying *inherent* obstacles against robust classification when the instances are drawn from the uniform distribution. We prove that for *any* learning problem P with input distribution $U_n$ (i.e., uniform over the hypercube) and for any classifier $h$ for P with a constant error $\mu$, the robustness of $h$ to adversarial perturbations (in Hamming distance) is at most $O(\sqrt{n})$. We also show that by the same amount of $O(\sqrt{n})$ perturbations *in the worst case*, one can increase the risk to 0.99. Table 1 lists some numerical examples.

Table 1: Each row focuses on the number of tampered bits to achieve its stated goal. The second column shows results using direct calculations for specific dimensions. The third column shows that these results are indeed achieved in the limit, and the last column shows bounds proved for all $n$.

| Adversarial goals | Types of bounds | | |
|---|---|---|---|
| | $n = 10^3, 10^4, 10^5$ | $n \mapsto \infty$ | all $n$ |
| From initial risk 0.01 to 0.99 | $\approx 2.34\sqrt{n}$ | $< 2.34\sqrt{n}$ | $< 3.04\sqrt{n}$ |
| From initial risk 0.01 to 0.50 | $\approx 1.17\sqrt{n}$ | $< 1.17\sqrt{n}$ | $< 1.52\sqrt{n}$ |
| Robustness for initial risk 0.01 | $\approx 1.17\sqrt{n}$ | $< 1.17\sqrt{n}$ | $< 1.53\sqrt{n}$ |

To prove results above, we apply the isoperimetric inequality of [26, 16] to the error region of the classifier $h$ and the ground truth $c$. In particular, it was shown in [16, 26] that the subsets of the hypercube with minimum "expansion" (under Hamming distance) are Hamming balls. This fact enables us to prove our bounds on the risk. We then prove the bounds on robustness by proving a general connection between risk and robustness that might obe of independent interest. Using the central limit theorem, we sharpen our bounds for robustness and obtain bounds that closely match the bounds that we also obtain by direct calculations (based on the isoperimetric inequalities and picking Hamming balls as error region) for specific values of dimension $n = 10^3, 10^4, 10^5$.

**Full version.** All proofs could be found in the full version of the paper[2], which also includes results related to the adversarial risk of monotone conjunctions, complementing the picture of Section 3.

## 2   General Definitions of Risk and Robustness for Adversarial Perturbations

**Notation.**   We use calligraphic letters (e.g., $\mathcal{X}$) for sets and capital non-calligraphic letters (e.g., $D$) for distributions. By $x \leftarrow D$ we denote sampling $x$ from $D$. In a classification problem $\mathsf{P} = (\mathcal{X}, \mathcal{Y}, \mathcal{D}, \mathcal{C}, \mathcal{H})$, the set $\mathcal{X}$ is the set of possible *instances*, $\mathcal{Y}$ is the set of possible *labels*, $\mathcal{D}$ is a set of distributions over $\mathcal{X}$, $\mathcal{C}$ is a class of *concept* functions, and $\mathcal{H}$ is a class of *hypotheses*, where any $f \in \mathcal{C} \cup \mathcal{H}$ is a mapping from $\mathcal{X}$ to $\mathcal{Y}$. An *example* is a *labeled instance*. We did not state the loss function explicitly, as we work with classification problems, however all main three definitions of this section directly extend to arbitrary loss functions. For $x \in \mathcal{X}, c \in \mathcal{C}, D \in \mathcal{D}$, the *risk* or *error* of a hypothesis $h \in \mathcal{H}$ is the expected (0-1) loss of $(h, c)$ with respect to $D$, namely $\mathsf{Risk}(h, c, D) = \Pr_{x \leftarrow D}[h(x) \neq c(x)]$. We are usually interested in learning problems with a fixed distribution $\mathcal{D} = \{D\}$, as we are particularly interested in robustness of learning under the uniform distribution $U_n$ over $\{0, 1\}^n$. Note that since we deal with negative results, fixing the distribution only makes our results stronger. As a result, whenever $\mathcal{D} = \{D\}$, we omit $D$ from the risk notation and simply write $\mathsf{Risk}(h, c)$. We usually work with problems $\mathsf{P} = (\mathcal{X}, \mathcal{Y}, D, \mathcal{C}, \mathcal{H}, \mathbf{d})$ that include a metric $\mathbf{d}$ over the instances. For a set $\mathcal{S} \subseteq \mathcal{X}$ we let $\mathbf{d}(x, \mathcal{S}) = \inf\{\mathbf{d}(x, y) \mid y \in \mathcal{S}\}$, and by $\mathcal{B}all_r(x) = \{x' \mid \mathbf{d}(x, x') \leq r\}$ we denote the ball of radius $r$ centered at $x$ under the metric $\mathbf{d}$. By HD we denote Hamming distance for pairs of instances from $\{0, 1\}^n$. Finally, we use the term *adversarial instance* to refer to an adversarially perturbed instance $x'$ of an originally sampled instance $x$ when the label of the adversarial example is either not known or not considered.

Below we present our formal definitions of adversarial risk and robustness. In all of these definitions we will deal with attackers who perturb the initial test instance $x$ into a *close* adversarial instance $x'$. We will measure how much an adversary can increase the *risk* by perturbing a given input $x$ into a *close* adversarial example $x'$. When to exactly call $x'$ a successful adversarial example is where these definitions differ. First we formalize the main definition that we use in this work based on adversary's ability to push instances to the error region.

**Definition 2.1** (Error-region risk and robustness). *Let* $\mathsf{P} = (\mathcal{X}, \mathcal{Y}, D, \mathcal{C}, \mathcal{H}, \mathbf{d})$ *be a classification problem (with metric* $\mathbf{d}$ *defined over instances* $\mathcal{X}$*).*

- **Risk.** *For any* $r \in \mathbf{R}_+, h \in \mathcal{H}, c \in \mathcal{C}$, *the* error-region risk *under* $r$-*perturbation is*

$$\mathsf{Risk}_r^{\mathrm{ER}}(h, c) = \Pr_{x \leftarrow D}[\exists x' \in \mathcal{B}all_r(x), h(x') \neq c(x')] \,.$$

*For* $r = 0$, $\mathsf{Risk}_r^{\mathrm{ER}}(h, c) = \mathsf{Risk}(h, c)$ *becomes the standard notion of risk.*

- **Robustness.** *For any* $h \in \mathcal{H}, x \in \mathcal{X}, c \in \mathcal{C}$, *the* error-region robustness *is the expected distance of a sampled instance to the error region, formally defined as follows*

$$\mathsf{Rob}^{\mathrm{ER}}(h, c) = \mathbb{E}_{x \leftarrow D}[\inf\{r \colon \exists x' \in \mathcal{B}all_r(x), h(x') \neq c(x')\}] \,.$$

Definition 2.1 requires the adversarial instance $x'$ to be *misclassified*, namely, $h(x') \neq c(x')$. So, $x'$ clearly belongs to the error region of the hypothesis $h$ compared to the ground truth $c$. This definition is implicit in the work of [14]. In what follows, we compare our main definition above with previously proposed definitions of adversarial risk and robustness found in the literature and discuss when they are (and when they are not) equivalent to Definition 2.1. Figure 1 summarizes the differences between the three main definitions that have appeared in the literature, where we distinguish cases by comparing the classifier's prediction $h(x')$ at the new point $x'$ with either of $h(x), c(x),$ or $c(x')$, leading to three different definitions.

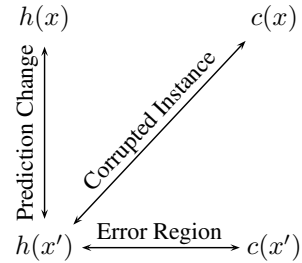

Figure 1: The three main definitions based on what $h(x')$ is compared with.

[2] See https://arxiv.org/abs/1810.12272.

**Definitions based on hypothesis's prediction change (PC risk and robustness).** Many works, including the works of [32, 11] use a definition of robustness that compares classifier's prediction $h(x')$ with the prediction $h(x)$ on the original instance $x$. Namely, they require $h(x') \neq h(x)$ rather than $h(x') \neq c(x')$ in order to consider $x'$ an adversarial instance. Here we refer to this definition (that does not depend on the ground truth $c$) as *prediction-change* (PC) risk and robustness (denoted as $\mathsf{Risk}_r^{\mathrm{PC}}(h)$ and $\mathsf{Rob}^{\mathrm{PC}}(h)$). We note that this definition captures the error-region risk and robustness if we *assume* the initial correctness (i.e., $h(x) = c(x)$) of classifier's prediction on all $x \leftarrow X$ and "truth proximity", i.e., that $c(x) = c(x')$ holds for all $x'$ that are "close" to $x$. Both of these assumptions are valid in some natural scenarios. For example, when input instances consist of images that look similar to humans (if used as the ground truth $c(\cdot)$) and if $h$ is also correct on the original (non-adversarial) test examples, then the two definitions (based on error region or prediction change) coincide. But, these assumptions do not hold in *in general*.

**Definitions based on the notion of corrupted instance (CI risk and robustness).** The works of [21, 12, 13, 1] study the robustness of learning models in the presence of *corrupted inputs*. A more recent framework was developed in [20, 29] for modeling risk and robustness that is inspired by robust optimization [3] (with an underlying metric space) and model adversaries that corrupt the the original instance in (exponentially more) ways. When studying adversarial perturbations using corrupted instances, we define adversarial risk by requiring the adversarial instance $x'$ to satisfy $h(x') \neq c(x)$. The term "corrupted instance" is particularly helpful as it emphasizes on the fact that the goal (of the classifier) is to find the *true* label of the *original* (uncorrupted) instance $x$, while we are only given a corrupted version $x'$. Hence, we refer to this definition as the *corrupted instance* (CI) risk and robustness and denote them by $\mathsf{Risk}_r^{\mathrm{CI}}(h, c)$ and $\mathsf{Rob}^{\mathrm{CI}}(h, c)$. The advantage of this definition compared to the prediction-change based definitions is that here, we no longer need to assume the initial correctness assumption. Namely, only if the "truth proximity" assumption holds, then we have $c(x) = c(x')$ which together with the condition $h(x') \neq c(x)$ we can conclude that $x'$ is indeed misclassified. However, if small perturbations can change the ground truth, $c(x')$ can be different from $c(x)$, in which case, it is no long clear whether $x'$ is misclassified or not.

**Stronger definitions of risk and robustness with more restrictions on adversarial instance.** The corrupted-input definition requires an adversarial instance $x'$ to satisfy $h(x') \neq c(x)$, and the error-region definition requires $h(x') \neq c(x')$. What if we require *both* of these conditions to call $x'$ a true adversarial instance? This is indeed the definition used in the work of Suggala et al. [31], though more formally in their work, they subtract the original risk (without adversarial perturbation) from the adversarial risk. This definition is certainly a *stronger* guarantee for the adversarial instance. As this definition is a hybrid of the error-region and corrupted-instance definitions, we do not make a direct study of this definition and only focus on the other three definitions described above.

**How about when the classifier $h$ is 100% correct?** We emphasize that when $h$ happens to be the same function as $c$, (the error region) Definition 2.1 implies $h$ has zero *adversarial* risk and infinite adversarial robustness $\mathsf{Rob}^{\mathrm{ER}}(h, c) = \infty$. This is expected, as there is no way an adversary can perturb any input $x$ into a misclassified $x'$. However, both of the definitions of risk and robustness based on prediction change [32] and corrupted instance [21, 20] could compute large risk and small robustness for such $h$. In fact, in a recent work [33] it is shown that for definitions based on corrupted input, correctness might be *provably at odds* with robustness in some cases. Therefore, even though all these definitions could perhaps be used to approximate the risk and robustness when we do not have access to the ground truth $c'$ on the new point $x'$, in this work we separate the *definition* of risk and robustness from how to compute/approximate them, so we will use Definition 2.1 by default.

## 3 A Comparative Study through Monotone Conjunctions

In this section, we compare the risk and robustness under the three definitions of Section 2 through a study of monotone conjunctions under the uniform distribution. Namely, we consider adversarial perturbations of truth assignments that are drawn from the uniform distribution $U_n$ over $\{0, 1\}^n$ when the concept class contains monotone conjunctions. As we will see, these definitions diverge in this natural case. Below we fix the setup under which all the subsequent results are obtained.

**Problem Setup 1.** *Let $\mathcal{C}_n$ be the concept class of all monotone conjunctions formed by at least one and at most $n$ Boolean variables. The target concept (ground truth) $c$ that needs to be learned is*

*drawn from $\mathcal{C}_n$. Let the hypothesis class be $\mathcal{H} = \mathcal{C}_n$ and let $h \in \mathcal{H}$ be the hypothesis obtained by a learning algorithm after processing the training data. With $|h|$ and $|c|$ we denote the size of $h$ and $c$ respectively; that is, number of variables that $h$ and $c$ contain.[3] Now let,*

$$c = \bigwedge_{i=1}^{m} x_i \wedge \bigwedge_{k=1}^{u} y_k \qquad \text{and} \qquad h = \bigwedge_{i=1}^{m} x_i \wedge \bigwedge_{\ell=1}^{w} z_\ell. \qquad (1)$$

*We will call the variables that appear both in $h$ and $c$ as* mutual, *the variables that appear in $c$ but not in $h$ as* undiscovered, *and the variables that appear in $h$ but not in $c$ as* wrong *(or* redundant*). Therefore in (1) we have $m$ mutual variables, $u$ undiscovered and $w$ wrong. We denote the error region of a hypothesis $h$ and the target concept $c$ with $\mathcal{E}(h,c)$.*

*That is, $\mathcal{E}(h,c) = \{x \in \{0,1\}^n \mid h(x) \neq c(x)\}$. The probability mass of the error region between $h$ and $c$, denoted by $\mu$, under the uniform distribution $U_n$ over $\{0,1\}^n$ is then,*

$$\Pr_{x \leftarrow U_n}[x \in \mathcal{E}(h,c)] = \mu = (2^w + 2^u - 2) \cdot 2^{-m-u-w}. \qquad (2)$$

*In this problem setup we are interested in computing the* adversarial risk *and* robustness *that attackers can achieve when instances are drawn from the uniform distribution $U_n$ over $\{0,1\}^n$.*

**Remark 3.1.** *Note that $\mu$ is a variable that depends on the particular $h$ and $c$.*

Using the Problem Setup 1, in what follows we compute the adversarial robustness that an arbitrary hypothesis has against an arbitrary target using the *error region (ER)* definition that we advocate in contexts where the perturbed input is supposed to be misclassified and do the same calculations for adversarial risk and robustness that are based on the definitions of *prediction change (PC)* and *corrupted instance (CI)*. The important message is that the adversarial robustness of a hypothesis based on the ER definition is $\Theta(\min\{|h|,|c|\})$, whereas the adversarial robustness based on PC and CI is $\Theta(|h|)$. In the full version of the paper we also give theorems (that have similar flavor) for calculating the adversarial risk based on the three main definitions (ER, PC, CI).

**Theorem 3.2.** *Consider the Problem Setup 1. Then, if $h = c$ we have $\mathsf{Rob}^{\mathrm{ER}}(h,c) = \infty$, while if $h \neq c$ we have $\min\{|h|,|c|\}/16 \leq \mathsf{Rob}^{\mathrm{ER}}(h,c) \leq 1 + \min\{|h|,|c|\}$.*

**Theorem 3.3.** *Consider the Problem Setup 1. Then, $\mathsf{Rob}^{\mathrm{PC}}(h) = |h|/2 + 2^{-|h|}$.*

**Theorem 3.4.** *Consider the Problem Setup 1. Then, $|h|/4 < \mathsf{Rob}^{\mathrm{CI}}(h,c) < |h| + 1/2$.*

### 3.1 Experiments for the Expected Values of Adversarial Robustness

In this part, we complement the theorems that we presented earlier with experiments. This way we are able to examine how some popular algorithms behave under attack, and we explore the extent to which the generated solutions of such algorithms exhibit differences in their (adversarial) robustness on average against various target functions drawn from the class of monotone conjunctions.

The first algorithm is the standard Occam algorithm that starts from the full conjunction and eliminates variables from the hypothesis that contradict the positive examples received; this algorithm is known as FIND-S in [22] but has appeared without a name earlier by Valiant in [34] and its roots are at least as old as in [6]. The second algorithm is the SWAPPING ALGORITHM from the framework of evolvability [35]. This algorithm searches for an $\varepsilon$-optimal solution among monotone conjunctions that have at most $\lceil \lg(3/(2\varepsilon)) \rceil$ variables in their representation using a local search method where hypotheses in the neighborhood are obtained by swapping in and out some variable(s) from the current hypothesis; we follow the analysis that was used in [10] and is a special case of [9].

In each experiment, we first learn hypotheses by using the algorithms under $U_n$ against different target sizes. For both algorithms, during the learning process, we use $\varepsilon = 0.01$ and $\delta = 0.05$ for the learning parameters. We then examine the robustness of the generated hypotheses by drawing examples again from the uniform distribution $U_n$ as this is the main theme of this paper. In particular, we test against the 30 target sizes from the set $\{1, 2, \ldots, 24, 25, 30, 50, 75, 99, 100\}$. For each such target size, we plot the average value, over 500 runs, of the robustness of the learned hypothesis that

we obtain. In each run, we repeat the learning process using a random target of the particular size as well as a fresh training sample and subsequently estimate the robustness of the learned hypothesis by drawing another $10,000$ examples from $U_n$ that we violate (depending on the definition). The dimension of the instances is $n = 100$.

Figure 2 presents the values of the three robustness measures for the case of FIND-S. In the full version of the paper we provide more details on the algorithms and more information regarding our experiments. The message is that the adversarial robustness that is based on the definitions of *prediction change* and *corrupted instance* is more or less the same, whereas the adversarial robustness based on the *error region* definition may obtain wildly different values compared to the other two.

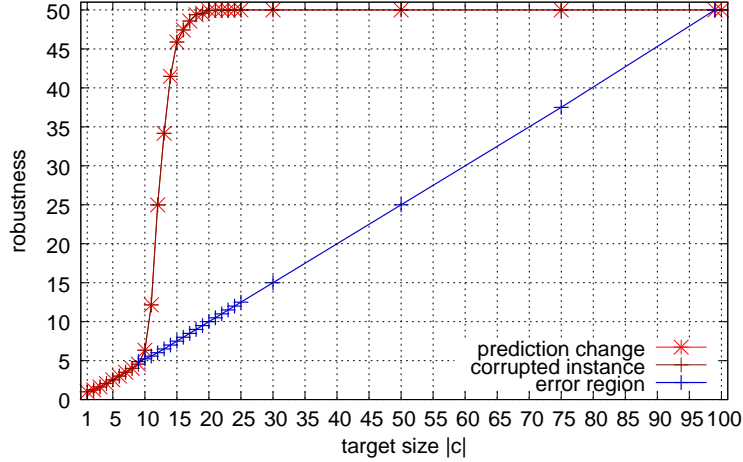

Figure 2: Experimental comparison of the different robustness measures. The values for PC and CI almost coincide and they can hardly be distinguished. The value for ER robustness is completely different compared to the other two. Note that ER robustness is $\infty$ when the target size $|c|$ is in $\{1, \ldots, 8\} \cup \{100\}$ and for this reason only the points between 9 and 99 are plotted. When $|c| \geq 20$, almost always the learned hypothesis is the initialized full conjunction. The reason is that positive examples are very rare and our training set contains none. As a result no variable is eliminated from the initialized hypothesis $h$ (full conjunction). Hence, when $|c| \geq 20$ we see that PC and CI robustness is about $\max\{|h|, |c|\}/2 = |h|/2$, whereas ER is roughly $\min\{|h|, |c|\}/2 = |c|/2$.

## 4   Inherent Bounds on Risk and Robustness for the Uniform Distribution

In this section, we state our main theorems about *error region* adversarial risk and robustness of arbitrary learning problems whose instances are distributed uniformly over the $n$-dimension hypercube $\{0, 1\}^n$. The proofs of the theorems below are available in the full version of the paper.

We first define a useful notation for the size of the (partial) Hamming balls.

**Definition 4.1.** *For every* $n \in \mathbb{N}$ *we define the (partial) "Hamming Ball Size" function* $\mathsf{BSize}_n \colon [n] \times [0, 1) \to [0, 1)$ *as follows*

$$\mathsf{BSize}_n(k, \lambda) = 2^{-n} \cdot \left( \sum_{i=0}^{k-1} \binom{n}{i} + \lambda \cdot \binom{n}{k} \right).$$

*Note that this function is a bijection and we use* $\mathsf{BSize}^{-1}(\cdot)$ *to denote its inverse. When $n$ is clear from the context, we will simply use* $\mathsf{BSize}(\cdot, \cdot)$ *and* $\mathsf{BSize}^{-1}(\cdot)$ *instead.*

The following theorem, gives a general lower bound for the adversarial risk of any classification problem for uniform distribution $U_n$ over the hypercube $\{0, 1\}^n$, depending on the original error.

**Theorem 4.2.** *Suppose* $\mathsf{P} = (\{0, 1\}^n, \mathcal{Y}, U_n, \mathcal{C}, \mathcal{H}, \mathsf{HD})$ *is a classification problem. For any* $h \in \mathcal{H}, c \in \mathcal{C}$ *and* $r \in \mathbb{N}$, *let* $\mu = \mathsf{Risk}(h, c) > 0$ *be the original risk and* $(k, \lambda) = \mathsf{BSize}^{-1}(\mu)$ *be a function of the original risk. Then, the error-region adversarial risk under $r$-perturbation is at least*

$$\mathsf{Risk}_r^{\mathrm{ER}}(h, c) \geq \mathsf{BSize}(k + r, \lambda).$$

The following corollary determines an asymptotic lower bound for risk based on Theorem 4.2.

**Corollary 4.3** (Error-region risk for all $n$). *Suppose* $\mathsf{P} = (\{0,1\}^n, \mathcal{Y}, U_n, \mathcal{C}, \mathcal{H}, \mathsf{HD})$ *is a classification problem. For any hypothesis $h, c$ with risk $\mu \in (0, \frac{1}{2}]$ in predicting a concept function $c$, we can increase the risk of $(h, c)$ from $\mu \in (0, \frac{1}{2}]$ to $\mu' \in [\frac{1}{2}, 1]$ by changing at most*

$$r = \sqrt{\frac{-n \cdot \ln \mu}{2}} + \sqrt{\frac{-n \cdot \ln(1 - \mu')}{2}}$$

*bits in the input instances. Namely, by using the above $r$, we have $\mathsf{Risk}_r^{\mathrm{ER}}(h, c) \geq \mu'$. Also, to increase the error to $\frac{1}{2}$ we only need to change at most $r' = \sqrt{\frac{-n \cdot \ln(\mu)}{2}}$ bits.*

**Example.** Corollary 4.3 implies that for classification tasks over $U_n$, by changing at most $3.04\sqrt{n}$ number of bits in each example we can increase the error of an hypothesis from $1\%$ to $99\%$. Furthermore, for increasing the error just to $0.5$ we need half of the number of bits, which is $1.52\sqrt{n}$.

Also, the corollary bellow, gives a lower bound on the limit of adversarial risk when $n \mapsto \infty$. This lower bound matches the bound we have in our computational experiments.

**Corollary 4.4** (Error-region risk for large $n$). *Let $\mu \in (0, 1]$ and $\mu' \in (\mu, 1]$ and $\mathsf{P} = (\{0,1\}^n, \mathcal{Y}, U_n, \mathcal{C}, \mathcal{H}, \mathsf{HD})$ be a classification problem. Then for any $h \in \mathcal{H}, c \in \mathcal{C}$ such that $\mathsf{Risk}(h, c) \geq \mu$ we have $\mathsf{Risk}_r(h, c) \geq \mu'$ for*

$$r \approx \sqrt{n} \cdot \frac{\Phi^{-1}(\mu') - \Phi^{-1}(\mu)}{2} \quad \text{when } n \mapsto \infty$$

*where $\Phi$ is the CDF of the standard normal distribution.*

**Example.** Corollary 4.4 implies that for classification tasks over $U_n$, when $n$ is large enough, we can increase the error from $1\%$ to $99\%$ by changing at most $2.34\sqrt{n}$ bits, and we can we can increase the error from $1\%$ to $50\%$ by changing $1.17\sqrt{n}$ bits in test instances.

The following theorem shows how to upper bound the adversarial *robustness* using the original risk.

**Theorem 4.5.** *Suppose* $\mathsf{P} = (\{0,1\}^n, \mathcal{Y}, U_n, \mathcal{C}, \mathcal{H}, \mathsf{HD})$ *is a classification problem. For any $h \in \mathcal{H}$ and $c \in \mathcal{C}$, if $\mu = \mathsf{Risk}(h, c)$ and $(k, \lambda) = \mathsf{BSize}^{-1}(\mu)$ depends on the original risk, then the error-region robustness is at most*

$$\mathsf{Rob}^{\mathrm{ER}}(h, c) \leq \sum_{r=0}^{n-k+1} (1 - \mathsf{BSize}(k + r, \lambda)).$$

Following, using Theorem 4.5, we give an asymptotic lower bound for robustness .

**Corollary 4.6.** *Suppose* $\mathsf{P} = (\{0,1\}^n, \mathcal{Y}, U_n, \mathcal{C}, \mathcal{H}, \mathsf{HD})$ *is a classification problem. For any hypothesis $h$ with risk $\mu \in (0, \frac{1}{2}]$, we can make $h$ to give always wrong answers by changing $r = \sqrt{-n \cdot \ln \mu / 2} + \mu \cdot \sqrt{n/2}$ number of bits on average. Namely, we have*

$$\mathsf{Rob}^{\mathrm{ER}}(h, c) \leq \sqrt{\frac{-n \cdot \ln \mu}{2}} + \mu \cdot \sqrt{\frac{n}{2}}.$$

*And the following Corollary gives a lower bound on the robustness in limit.*

**Corollary 4.7.** *For any $\mu \in (0, 1]$, classification problem $\mathsf{P} = (\{0,1\}^n, \mathcal{Y}, U_n, \mathcal{C}, \mathcal{H}, \mathsf{HD})$, and any $h \in \mathcal{H}, c \in \mathcal{C}$ such that $\mathsf{Risk}(h, c) \geq \mu$, we have*

$$\mathsf{Rob}^{\mathrm{ER}}(h, c) \leq \frac{\Phi^{-1}(\mu)}{2} \cdot \sqrt{n} + \mu \cdot \sqrt{\frac{\pi \cdot n}{8}} \quad \text{when} \quad n \mapsto \infty,$$

*where $\Phi$ is the CDF of the standard normall distribution.*

**Example.** By changing $1.53\sqrt{n}$ number of bits *on average* we can increase the error of an hypothesis from $1\%$ to $100\%$. Also, if $n \mapsto \infty$, by changing only $1.17\sqrt{n}$ number of bits *on average* we can increase the error from $1\%$ to $100\%$.

## Footnotes

*Authors have contributed equally.

†Supported by NSF CAREER CCF-1350939 and University of Virginia SEAS Research Innovation Award.

[1]For example, http://rodrigob.github.io/are_we_there_yet/build/ has a summary of state-of-the-art results.

[3] For example, $h_1 = x_1 \wedge x_5 \wedge x_8$ is a monotone conjunction of three variables in a space where we have $n \geq 8$ variables and $|h_1| = 3$.

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
