[Reviews · NeurIPS 2018]

Reviewer 1



The paper studies the adversarial examples phenomenon from the theoretical perspective with a focus on discrete distributions. The authors make two main contributions: 1. They introduce a new robustness measure called "error region robustness" and compare it to other definitions of robustness (the proposed measure is not specific to discrete distributions). The main difference to prior work is that the proposed definition does not count an example as robustly misclassified if the small perturbation actually changes the correct class. 2. They study learning problems under the uniform distribution on the hypercube and show that *any* classifier with a given standard risk will have a certain amount of adversarial / robust risk. The argument is based on isoperimetry results for the hypercube. I think both points are relevant and the theoretical arguments in the main paper seem correct. Compared to prior work, the novelty is somewhat limited however: - While their error region robustness is indeed interesting, it only differs from prior definitions of robustness when small changes to an example can change the correct class. In adversarial examples, the focus is typically on small perturbations that do not change the correct class. In this regime, the proposed definition and the definition in prior work coincide. - At a high level, the isoperimetry argument relating standard accuracy to robust accuracy is similar to the work of Gilmer et al. (cited by the authors) for a continuous distribution. The authors mention twice that they give the first results on adversarial robustness for a discrete distribution. It seems however that the work of Schmidt et al. (also cited) already establishes robustness results for a distribution on the hypercube. It would be helpful if the authors could comment on this in the rebuttal. But it is worth noting that as far as I know, the hypercube isoperimetry argument for the uniform distribution has not appeared before in the context of adversarial examples. Another area of improvement is that the authors could discuss in more detail what their results mean for the image classification setting that is common in the adversarial examples literature. For instance, it is not clear how restrictive the assumption of a uniform distribution on the hypercube is (in terms of images, this would correspond to fully random images). As far as I can tell, the theoretical results crucially rely on working with a uniform distribution. While simple models are certainly a good starting point for theoretical studies, it would be good to discuss the connections to empirical phenomena. Finally, it seems that the error region notion of robustness might have counter-intuitive behavior in regimes where the Bayes error is strictly greater than 0. In that case, the error-region risk of the Bayes-optimal classifier will be 0 while the "standard" risk is non-zero (i.e., the robust risk is less than the "standard" risk). The prior notions of robust risk do not have this shortcoming. It would be helpful if the authors could comment on this in their rebuttal and potentially include a discussion about this in their paper. Overall, my current assessment is a weak accept since the two main points are important and there is currently a lack of theoretical results regarding adversarial robustness. Minor comments: - There are some formatting issues. The abstract is too long and the paper was submitted using the "preprint" option of the NIPS LaTeX package. - On Page 3, it would be helpful to remind the reader what a monotone conjunction is. - The second paragraph from the bottom of Page 3 seems out of context as it mentions several terms that are not defined yet (OT, PC, h, c, etc.). - The abstract contains two sentences that seem ungrammatical: * "Our experimental calculations point for range of [...]" * "[...] previously proposed definitions have risk and robustness e.g., by pushing [...]" - Near the bottom of Page 2, should the bound 2.15 sqrt(n) depend on mu from the previous sentence? Or is it for a specific mu? - Page 4: "We did not state the loss function explicitly [...]" is missing a "." at the end. - In Definition 3.6, would it be easier to use |A| (cardinality of A) instead of V(A) since we are working with a uniform distribution over the hypercube? The paper uses |A| in other places such as Lemmas 3.7 and 3.8. - What is p_i in the proof of Lemma 3.9? In particular, it is not clear what the index i refers to. - In Equation (2) on Page 8, should it be (2^w + 2^u - 1)? - The sentence before Theorem 3.10 refers to a discussion in Section D of the appendix, but the section seems to contain only proofs. Is this intentional? - Appendix B is empty. - Appendix D.7 mentions "10,00" random examples. The use of "," here is non-standard.

Reviewer 2



The paper studies the robustness of classifiers to adversarial perturbations in the (simple) setting where all the input features are binary. It provides upper bounds on the number of features that the adversary must flips to make the prediction irrelevant. The paper also provides empirical comparison of their measure of robustness with other measures of the literature. The proposed measure seems more appropriate, as it measures the risk compare to the ground truth As a reader unfamiliar with the adversarial attacks topic, I enjoyed the fact that the paper is really clearly written. It is easy to follow without being trivial, and provides both theoretical and numerical results. One may complains that the simplicity of the setting clash with most nowadays machine learning applications. From my point of view, the less realistic assumption is that the data distribution is uniform over the input space. I would like the authors to comment on this point. Can this assumption be relaxed? That being said, the fact that the problem is well circumvented allows to study it with rigor. Hence, it appears to me to be the kind of work that may be the starting point to study more complex frameworks. Other comments and typos: - Page 3: Please provide details about the experiments of Table 1 - Page 5, Line 5: missing reference after "robust optimization" - Page 5, Line 7: "definitoin" (typo) - Theorem 2: Please state that "HD" stands for the Hamming distance metric, and thus r is an integer value

Reviewer 3



This paper proposes a new definition for “robustness” of the model, and analyzes its lower bound for discrete uniform distribution over the corners of the hypercube (all binary vectors). The results are presented clearly and the techniques used in this paper are interesting. In a nutshell, based on a isoperimetric inequality for hypercube with Hamming distance, this paper calculates a lower bound for the expansion speed of the boundary of any set, which leads to the lower bounds. The new definition of “robustness” in the paper seems reasonable. Its relations to the previous definitions are also discussed in the paper. The only concern I have of this paper is about the motivation/significance of the study. 1. The authors justify this research in the introduction, emphasizing the importance of studying discrete distributions in the literature. But the papers referred there are all about DNF. I understand the fundamentalness of the DNF problems. But some more discrete examples would give much better justification. 2. It is also not clear why studying “robustness” is important, even in the context of DNF. Also, in section 3.3, only results for monotone conjunctions are presented (Not quite yet about DNF). Could the authors propose some learning settings, where the robustness of learning a monotone conjunctions plays an essential role? Other comments: 1. The proof of Theorem 3.2 seems to miss the case when E is empty. Also in Theorem 3.2, Risk(h,c) is not yet defined in the paper 2. In section 1.1 about the monotone conjunction part, PC and OT should be introduced before they are used. PC is also not consistent with PCh on page 4. 3. On page 7 in the proof of Theorem 3.2, Line 4 and 5, K_E -> k_E. It also misses a denominator 2^n for the last term. 4. In Theorem 3.10/3.12, \rho(h,U_n,c) -> \rho(h ,c) Disclaimer: I didn’t read the proofs for Section 3.3. ==Update after rebuttal ========================= 1. I don't think the rebuttal addresses the significance problem raised in my review. But I change my score to 6 for its rigorous manner in the adversarial robustness literature. 2. I misunderstood question 4 in my original review, thus it is updated to 2 now. My apologies.